# Major Bleeding Predictors in Patients with Left Atrial Appendage Closure: The Iberian Registry II

**DOI:** 10.3390/jcm9072295

**Published:** 2020-07-19

**Authors:** José Ramón López-Mínguez, Juan Manuel Nogales-Asensio, Eduardo Infante De Oliveira, Lino Santos, Rafael Ruiz-Salmerón, Dabit Arzamendi-Aizpurua, Marco Costa, Hipólito Gutiérrez-García, Jose Antonio Fernández-Díaz, Xavier Freixa, Ignacio Cruz-González, Raúl Moreno, Andrés Íñiguez-Romo, Fernando Alfonso-Manterola

**Affiliations:** 1Cardiology Department, Interventional Cardiology Section, Hospital Universitario de Badajoz, 06080 Badajoz, Spain; juanmanogales@yahoo.es; 2Cardiology Department, Interventional Cardiology Section, Hospital de Santa María, 1649-028 Lisbon, Portugal; e.infante.de.oliveira@gmail.com; 3Cardiology Department, Interventional Cardiology Section, Centro Hospitalario de Vila Nova de Gaia, 4430-999 Vila Nova de Gaia Oporto, Portugal; ljsantos30@gmail.com; 4Cardiology Department, Interventional Cardiology Section, Hospital Virgen de la Macarena, 41009 Seville, Spain; rjruizsalmeron@yahoo.es; 5Cardiology Department, Interventional Cardiology Section, Hospital Santa Creu i San Pau, 08041 Barcelona, Spain; dabitarza@gmail.com; 6Cardiology Department, Interventional Cardiology Section, Centro Hospitalar e Universitário de Coimbra, 3004-561 Coimbra, Portugal; marcocostacard@sapo.pt; 7Cardiology Department, Interventional Cardiology Section, Hospital Clínico de Valladolid, 47003 Valladolid, Spain; hggmaire@gmail.com; 8Cardiology Department, Interventional Cardiology Section, Hospital Puerta de Hierro, Majadahona, 28222 Madrid, Spain; joseantoniofer@gmail.com; 9Cardiology Department, Interventional Cardiology Section, Hospital Clínic de Barcelona, 08036 Barcelona, Spain; xavierfreixa@hotmail.com; 10Cardiology Department, Interventional Cardiology Section, Hospital Universitario de Salamanca, 37007 Salamanca, Spain; cruzgonzalez.ignacio@gmail.com; 11Cardiology Department, Interventional Cardiology Section, Hospital La Paz, 28046 Madrid, Spain; raulmorenog@hotmail.com; 12Cardiology Department, Interventional Cardiology Section, Hospital Álvaro Cunqueiro, 36213 Vigo, Pontevedra, Spain; Andres.Iniguez.Romo@sergas.es; 13Cardiology Department, Interventional Cardiology Section, Hospital La Princesa, IIS-IP, CIBER-CV, Universidad Autónoma de Madrid, 28006 Madrid, Spain; falf@hotmail.com

**Keywords:** atrial fibrillation, bleeding risk, age, left atrial appendage closure

## Abstract

Introduction and objective: Major bleeding events in patients undergoing left atrial appendage closure (LAAC) range from 2.2 to 10.3 per 100 patient-years in different series. This study aimed to clarify the bleeding predictive factors that could influence these differences. Methods: LAAC was performed in 598 patients from the Iberian Registry II (1093 patient-years; median, 75.4 years). We conducted a multivariate analysis to identify predictive risk factors for major bleeding events. The occurrence of thromboembolic and bleeding events was compared to rates expected from CHA2DS2-VASc (congestive heart failure, hypertension, age, diabetes, stroke history, vascular disease, sex) and HAS-BLED (hypertension, abnormal renal and liver function, stroke, bleeding, labile INR, elderly, drugs or alcohol) scores. Results: Cox regression analysis revealed that age ≥75 years (HR: 2.5; 95% CI: 1.3 to 4.8; *p* = 0.004) and a history of gastrointestinal bleeding (GIB) (HR: 2.1; 95% CI: 1.1 to 3.9; *p* = 0.020) were two factors independently associated with major bleeding during follow-up. Patients aged <75 or ≥75 years had median CHA2DS2-VASc scores of 4 (IQR: 2) and 5 (IQR: 2), respectively (*p* < 0.001) and HAS-BLED scores were 3 (IQR: 1) and 3 (IQR: 1) for each group (*p* = 0.007). Events presented as follow-up adjusted rates according to age groups were stroke (1.2% vs. 2.9%; HR: 2.4, *p* = 0.12) and major bleeding (3.7 vs. 9.0 per 100 patient-years; HR: 2.4, *p* = 0.002). Expected major bleedings according to HAS-BLED scores were 6.2% vs. 6.6%, respectively. In patients with GIB history, major bleeding events were 6.1% patient-years (HAS-BLED score was 3.8 ± 1.1) compared to 2.7% patients-year in patients with no previous GIB history (HAS-BLED score was 3.4 ± 1.2; *p* = 0.029). Conclusions: In this high-risk population, GIB history and age ≥75 years are the main predictors of major bleeding events after LAAC, especially during the first year. Age seems to have a greater influence on major bleeding events than on thromboembolic risk in these patients.

## 1. Introduction

Left atrial appendage closure (LAAC) is a therapeutic option for patients with a high bleeding risk even in the absence of anticoagulant (AC) treatment after LAAC, and its use has been supported by the increasing body of evidence obtained from several studies and registries [1].

Comparison of results from the series of patients who underwent LAAC may show variations in the percentage of events during follow-up. In spite of some consistency in the relative reduction of stroke (60–80% of CHA2DS2-VASc (congestive heart failure, hypertension, age, diabetes, stroke history, vascular disease, sex) scores), there was a higher variability in bleeding events among the series from the first year of follow-up, with major bleeding events ranging from 2.2% to 10% per year [2,3,4].

We searched for variables that might be independent predictors for major bleeding events at follow-up. We based our analysis on the Iberian Registry II [5].

## 2. Methods

### Patients and Procedures

Five hundred and ninety-eight patients from the Iberian Registry II referred for LAAC were recruited from 13 hospitals across the Iberian Peninsula (10 from Spain and 3 from Portugal) between 2 March 2009 and 18 December 2015 [5]. These were the set of patients prospectively included in the Iberian Registry I who are continuing long-term follow-up, plus additional patients successively included up to the end of the date set for end of recruitment. Inclusion criteria were one or more of the following conditions: serious hemorrhage during anticoagulant therapy, prior disease or clinical event that contraindicated oral anticoagulants (OACs) or repeated failure to adequately control INR, and hematologist indication to suspend anticoagulation therapy. Exclusion criteria were malignancy, life expectancy less than one year and refusal to provide informed consent for this study.

LAAC indication was as follows: stroke under OAC therapy 6.2%, previous bleeding 73.7%, high risk of bleeding 14.2% and other (poorly controlled INR, patient decision, etc.) 5.9%. Before LAAC, 74.8% and 25.2% of the patients were under OAC and antiplatelet therapy respectively.

The devices used were the Amplatzer^®^ Cardiac Plug (ACP) and its subsequent version, the Amulet^®^ (both from St. Jude Medical; Minneapolis, MN, USA), and the Watchman^®^ (Boston Scientific; Boston, MA, USA).

Thromboembolic and bleeding events were compared with those expected from CHA2DS2-VASc (congestive heart failure, hypertension, age, diabetes, stroke history, vascular disease, sex) and HAS-BLED (hypertension, abnormal renal and liver function, stroke, bleeding, labile INR, elderly, drugs or alcohol) scores in the overall sample [6,7]. Major bleeding events were defined according to VARC-2 classification [8].

The observed incidence of events (number of events during the follow-up period divided by the number of patients per year of follow-up) was calculated per patient and year of follow-up (number of patients at the beginning of the follow-up period multiplied by the mean time of follow-up of those patients expressed in years). The expected incidence of events in the sample was calculated as the mean of the individual risk of each patient. Each patient was assigned an individual risk according to a score of bleeding and ictus risk depending on his or her CHADS2 and HAS-BLED score, as indicated in the work by Friberg and colleagues in the Swedish Atrial Fibrillation cohort study. 

All subjects gave their informed consent for inclusion before they participated in the study. The study was conducted in accordance with the Declaration of Helsinki, and the protocol was approved by the Ethics Committee of Hospital Universitario de Badajoz (Project identification 5517).

## 3. Statistical Analysis

Quantitative variables are expressed as mean (±standard deviation (SD)) or median (interquartile range (IQR)). Categorical variables are expressed as absolute frequency and percentage. Categorical variables were compared using the χ2 test or Fisher’s exact test, and quantitative variables using the Student t-test or Wilcoxon test. Comparisons between rates of observed and expected events were evaluated using binomial tests. Event-free survival analysis was performed using the Kaplan–Meier method and Cox regression. Multivariate analysis (Cox regression) was performed to identify which variables might be independent predictors for bleeding events. Proportional-hazard assumption for Cox Regression was checked by use of Cox proportional hazards regression test with time-dependent covariates. All analyses were carried out using the SPSS statistical package, version 19.0.

All patients gave their consent authorizing the intervention and subsequent follow-up. The study protocol was approved by the hospital ethics committee and conforms to the ethical guidelines of the 1975 Declaration of Helsinki. More details of patients, work methods, variable definitions and statistical analyses were previously reported [5].

## 4. Results

In the Iberian Registry II, during a mean follow-up of 22.9 months, the observed events for stroke and major bleeding events according to CHA2DS2-VASc and HAS-BLED scores in the total population were 1.6% (vs. expected 8.5%) and 3.9% (vs. expected 6.4%), with a relative risk reduction (RRR) of 81% for stroke and 39% for major bleeding events. In patients monitored for more than 24 months (683 patient-years), stroke and bleeding frequencies were 1.5% and 2.6%, with RRRs of 82% and 59%, respectively.

In the univariate analysis, the variables that were associated with a higher rate of “major bleeding events” at follow-up were: age ≥ 75 years (HR: 2.8; 95% CI: 1.5–5.2; *p* = 0.002), gastrointestinal bleeding (GIB) history (HR: 2.3; 95% CI95: 1.2–4.3; *p* = 0.007) and the antecedent of hypertension (HR: 0.5; 95% CI: 0.3–1; *p* = 0.047). Multivariate analysis (Cox regression analysis) showed that the variables associated with “major bleeding events” during follow-up were only age ≥ 75 years (HR: 2.5; 95% CI: 1.3 to 4.8; *p* = 0.004) and GIB history (HR: 2.1; 95% CI: 1.1 to 3.9; *p* = 0.020).

In patients with previous bleeding, these occurred in 82.1% and 17.9% of patients under OAC and antiplatelet therapy, respectively. GIB accounted for 55% of previous major bleeding events and 82% of major bleeding events during follow-up. Most of GIB (28 of 35) took place during the first 12 months (25 of them in the first 6 months).

Patients were then divided into two different populations according to their age: <75 or ≥75 years (326 vs. 272 patients). Table 1 shows the main clinical variables between the two groups. The percentage of patients aged ≥ 75 years with a previous history of bleeding was 81.3%. In general, the percentage of patients with a history of intracranial hemorrhage (ICH) was lower (23.9% and 29.1% in older and younger groups; *p* = 0.14) compared to the percentage with a history of GIB or major bleeding events, which was even significantly higher in patients ≥75 years (GIB: 48.5% vs. 32.5%; *p* < 0.001; major bleeding: 53.3% vs. 39.9%; *p* = 0.001). There was also a higher percentage of patients with anemia and renal failure in the elderly group (Table 1). There were no significant differences with regard to the implant used in the procedure or in complications. Table 2 shows the following events presented as follow-up adjusted rates according to age group (<75 or ≥75 years): deaths, 3.9% vs. 11.8% (HR: 3; *p* < 0.001); stroke, 1.2% vs. 2.9% (HR: 2.4; *p* = 0.12); ICH, 1.2% vs. 0.2% (HR: 0.2; *p* = 0.09); GIB, 1.5% vs. 6.9% (OR: 4.6; *p* < 0.001) and major bleeding, 3.7 vs. 9.0 (HR: 2.4; *p* = 0.002) per 100 patient-years (corresponding to patients <75 vs. ≥75 years, respectively). A significant decrease in bleeding events occurred after 1 year of follow-up in both groups (0.5 and 2.9 per 100 patient-years for GIB (*p* = 0.045), and 0.4, and 1.9 per 100 patient-years (*p* = 0.018) for major bleeding, in younger and older patients, although patients aged ≥75 years continued to have more bleeding events) (Figure 1A,B). Figure 2 shows that survival rate with no GIB was significantly higher in patients aged <75 years compared to the elderly group.

Table 3 shows the rate of events per patient-year in patients aged ≥75 years compared to the rate expected by the risk scores. There were only small differences in stroke and ICH per patient-year between patients aged ≥75 or <75 years. In contrast, there was an increase in major bleeding events in patients aged ≥75 years, which did not occur for the group aged <75 years, in whom bleeding reduction was 40.3%. Most GIB events occurred in the first 6 months of follow-up in both groups. The two variables that were independently associated with the event “gastrointestinal bleeding” after multivariate analysis (Cox regression) were: age ≥75 years (HR: 3.0; 95% CI: 1.4 to 6,3; *p* = 0.004) and gastrointestinal bleeding (GIB) history (HR: 4.3; 95% CI: 2.0 to 9.4; *p* < 0.001).

Table 4 compares treatments at discharge between both age groups of patients, showing no significant differences. Table 5 divides patients in two groups based on new GIB or no new GIB events during follow-up and compares the main variables between the two subpopulations. We found significant differences in GIB history (74.3% vs. 37.7%, *p* < 0.001), HAS-BLED scores (3.8 ± 1.1 vs. 3.4 ± 1.2, *p* = 0.029), ICH (11.4% vs. 27.7%, *p* = 0.035) and death (25.7% vs. 12.6%, *p* = 0.027). Although patients with GIB history showed a higher death rate, multivariate analysis results showed that only age (HR: 1.08; 95% CI: 1.05 to 1.13; *p* < 0.001) and previous stroke (HR: 3.22; 95% CI: 1.64 to 6.34; *p* = 0.001) were predictive factors for death during follow-up.

Table 6 presents the main variables of LAAC patients with contraindications for OAC treatment, collected from different registries (data from > 500 patients).

## 5. Discussion

Our study shows that during follow-up, both age and major GIB history are the main predictors of major bleeding events and therefore both variables should be taken into account when making comparisons between bleeding percentages in the different series of patients. It is also important that results are interpreted according to the follow-up time, as bleeding rates are higher during the first year [9].

### 5.1. The Importance of GIB, Major Bleeding History and Follow-up Time

Only around 16%-17% of patients included in randomized trials of both non-vitamin K antagonist oral anticoagulant (NOAC) and LAAC with warfarin therapies have a previous history of bleeding [10]; conversely, in LAAC registries, major bleeding events range from 31% in the EWOLUTION Registry to 72% in the Amulet Registry, with the remainder having conditions associated with a high bleeding risk, such as severe anemia [2,5,11,12,13].

We show in Table 2 that GIBs represent a higher percentage of severe bleedings in older patients (≥75 years).

In the Amulet and II Iberian Registries, where we find higher HAS-BLED scores and higher percentages of patients with major bleeding history, major bleeding during follow-up rises to 10.3% and 5.4%, respectively, during the first year, in contrast to the results of the Multicentre and EWOLUTION studies, that range from 2.1% to 2.7% [2,9,11,12].

However, in the Iberian Registry II, bleeding events were reduced to 3.9% after two years of follow-up and to 2.6% after more than two years, which corresponded to a relative reduction of more than 21% (39% after two years and 58.7% after more than two years) [5].

Bleeding events, especially GIB, were higher during the first 6 months post-procedure, which is the time window when a higher percentage of patients were receiving dual antithrombotic treatment (therapy changed from two antiplatelet agents to only one after the first 3–6 months). It is interesting to note that bleeding reduction is less dramatic in patients with major bleeding history. The EWOLUTION study reported that after 2 years, patients with HAS-BLED scores <3 showed relative reductions of 50% (1.8% vs. expected 3.6%) compared to relative reductions of 41% in patients with HAS-BLED scores >3 (4.2% vs. expected 7.1%), and these figures were even lower in patients with a major bleeding history (30%) [14]. In our study, in patients with GIB history, major bleeding events were 6.1 per 100 patient-years (HAS-BLED score 3.8 ± 1.1), compared to 2.7 per 100 patient-years in patients with no previous GIB history (HAS-BLED score 3.4 ± 1.2; *p* = 0.029).

### 5.2. The Importance of Age

Age is a risk factor in the prediction of both stroke and bleeding as a whole [7]. Elderly patients receiving treatment with OAC present a high bleeding risk that ranges from 9% to 13%, and for that reason they are not well represented in NOAC randomized trials [15,16]. It is still debated whether age has more influence on major bleedings than on thromboembolic events, as several studies support opposite claims in this respect [17,18,19]. Thus, it was observed that even in patients who were able to take OAC in the ENGAGE AF-TIMI 48 study, age had a greater influence on major bleeding than on thromboembolic risk [17].

It is crucial to clarify if age has more impact on major bleeding risk or on thromboembolic events in NVAF patients undergoing left atrial appendage closure (LAAC) and with a history of frequent bleeding events. In our study, patients ≥75 years had higher HAS-BLED scores (3.5 ± 1.1) than patients <75 years (3.3 ± 1.2; *p* = 0.004), and bleeding events were 9 per 100 patient-years (26.7% increase) versus 3.7 per 100 patient-years in the younger group (40.3% decrease).

The question as to whether age is the main factor or a secondary factor responsible for the accumulation in the percentage of patients with bleeding history was clarified by the multivariate analysis, which showed that age and bleeding history were independent predictors of subsequent bleeding events, especially a history of GIB.

There are no specific studies on the importance of age in bleeding events in LAAC patients, although there are two published LAAC series from the Multicentre and EWOLUTION registries [4,20].

The study published by Freixa and colleagues, based on the Multicentre Registry with the AMPLATZER Cardiac Plug, compared two populations of patients under or over 75 years old [4] and found no differences in major bleeding events (1.7% vs. 2.6%; *p* = 0.54) during a mean follow-up of 16.8 months. In addition, in the series of Freixa and collaborators, the percentage of patients <75 or ≥75 years with a previous history of major bleeding was similar between the two groups (48.4% vs. 47.7%; *p* = 0.83), whereas in our study, the percentage was significantly different between the two groups and even higher in patients ≥75 years (39.9% vs. 53.3%; *p* = 0.001) [4,5].

Cruz-González and colleagues published a sub-analysis of the EWOLUTION Registry of patients undergoing LAAC, comparing patients <85 and ≥85 years old. Although the differences between younger and older patients were not statistically significant (due to a limited sample size), after a follow-up of 24 months the group aged >85 years presented higher rates of major or severe non-procedural bleeding events than the younger group (5.1 per 100 patient-years vs. 2.6 per 100 patient-years, or 7.5 vs. 4.3 respectively, if procedural severe bleedings were included) [20].

### 5.3. Post-Implantation Treatment is an Important Variable

Most of the patients included in our study underwent dual antiplatelet treatment (DAPT) for 3 months, and after that, treatment was reduced to only one antiplatelet agent (APT), which is the usual procedure in current studies. It is clear that post-interventional treatment is an important variable to take into account, but in all studies, with the exception of randomized series, patients present a high percentage of bleeding events of different natures and origins. This makes it difficult to standardize treatment guidelines, and doctors must make decisions based on the specific risk associated with each patient [14]. The analysis of this variable can be confusing as data from the Amulet Registry showed that patients that were not taking antiplatelet agents developed more bleeding events than the group that was medicated, reflecting the current trend to treat patients with lower bleeding risk instead of patients with a very high bleeding risk [2]. The EWOLUTION study showed that patients treated with DAPT presented more major bleeding events after 105 days than those who discontinued this treatment (3.5% vs. 1.1%) [14].

Our study has the limitations of any registry since it is not a randomized trial. However, patients cannot be randomized for ethical reasons. Nevertheless, our data reflect a very exhaustive collection of events in a highly complex population, and their comparison with expected outcomes according to the risk scores has been widely validated. Despite the difficulty of reaching a consensus regarding appropriate post-interventional antiplatelet treatment (generally DAPT for 3 months), our analyses took into consideration the duration of antiplatelet treatment when comparing patients with or without bleeds.

## 6. Conclusions

In our Iberian Registry II, 46% of patients referred for LAAC had previous major bleedings, mostly of gastrointestinal origin. A history of severe GIB is an independent predictor of new severe bleeding events during follow-up. The percentage of patients aged ≥75 years may also significantly influence the incidence of major bleeding events beyond that expected using the HAS-BLED score, especially due to the high frequency of GIB, as age appears to have greater influence on major bleeding than on thromboembolic risk in these patients. Despite this, after the first year, bleeding events fell significantly, although they continued to be higher than in the group aged < 75 years, in whom fewer bleeding complications were observed than expected from the HAS-BLED score. Efficacy in thromboembolic events remains very high, regardless of age, even from the first year.

### 6.1. What is Known about the Topic?

-LAAC is an effective therapeutic option for atrial fibrillation patients with a contraindication for the use of anticoagulants.-However, these patients present a high bleeding risk even in the absence of antiplatelet treatment.-Age influences the emergence of complications during follow-up of LAAC patients.

### 6.2. What does this Study add?

-This study shows that age has a greater influence on the occurrence of major bleedings than on thromboembolic events.-Our analysis also shows that GIB history is the main predictive factor of major bleeding events during the first year of follow-up after LAAC.-Differences in the rates of major bleeding events reported in different series of LAAC patients may be due to the number of patients ≥ 75 years and the percentage of patients with GIB history included in those series.

## Figures and Tables

**Figure 1 jcm-09-02295-f001:**
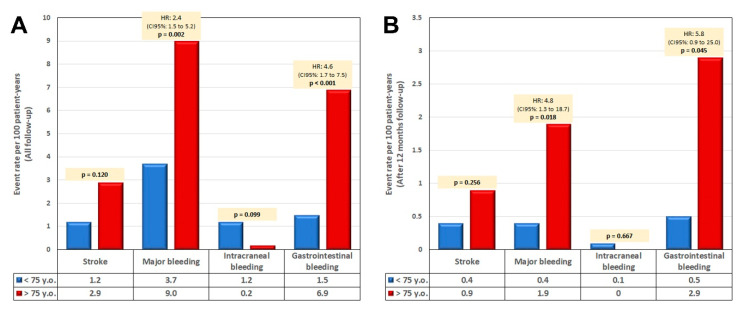
(**A**) Rate of events in patient-years according to age < or ≥75 years. (**B**) Rate of events in patients who completed the first year with no events. HR: hazard ratio; CI: confidence interval.

**Figure 2 jcm-09-02295-f002:**
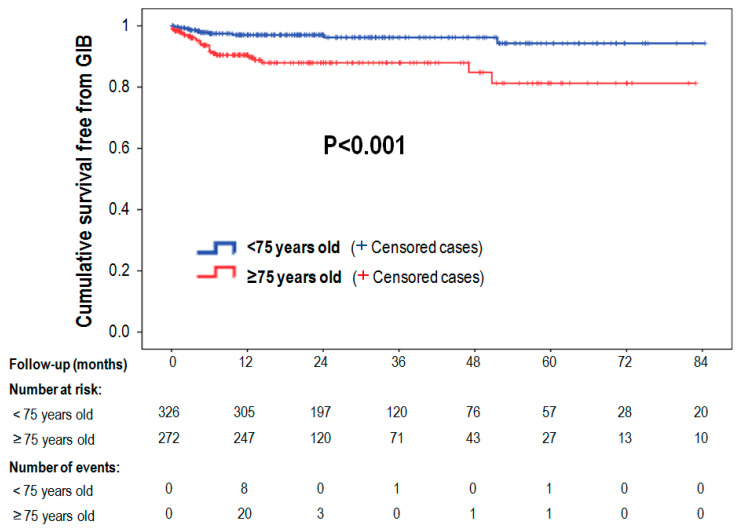
Cumulative survival free from gastrointestinal bleeding (GIB) is significantly higher in patients <75 years compared to patients ≥75 years.

**Table 1 jcm-09-02295-t001:** Clinical variables in the populations aged < 75 and ≥ 75 years.

	<75 (*n* = 326)	≥75 (*n* = 272)	*p*-Value
Age	67.3 ± 5.8	80.3 ± 3.5	<0.001
Female	112 (34.4%)	116 (42.6%)	0.038
Hypertension	255 (78.2%)	213 (78.3%)	0.979
Diabetes	106 (32.5%)	98 (36.0%)	0.367
Permanent AF	149 (45.7%)	159 (58.5%)	0.002
Previous stroke	111 (34.0%)	77 (28.3%)	0.132
Previous bleeding	170 (52.1%)	221 (81.3%)	<0.001
Previous ICH	95 (29.1%)	65 (23.9%)	0.149
Previous GI bleeding	106 (32.5%)	132 (48.5%)	<0.001
Previous major bleeding	130 (39.9%)	145 (53.3%)	0.001
CHA_2_DS_2_-VASc *	4 [2]	5 [2]	<0.001
HAS-BLED *	3 [1]	3 [1]	0.007
Anemia	63 (19.3%)	102 (37.5%)	<0.001
Renal failure	42 (12.9%)	85 (31.3%)	<0.001

AF: atrial fibrillation; GI: gastrointestinal; ICH: intracranial hemorrhage. * Median (IQR).

**Table 2 jcm-09-02295-t002:** Adjusted event rates per 100 patient-years.

	<75 (*n* = 326)	≥75 (*n* = 272)	HR (<75 vs. ≥75)	*p*-Value
Death	3.9	11.8	3.0	<0.001
Stroke	1.2	2.9	2.4	0.120
ICH	1.2	0.2	0.2	0.099
GI bleeding	1.5	6.9	4.6	<0.001
Major bleeding	3.7	9.0	2.4	0.002

GI: gastrointestinal; ICH: intracranial hemorrhage.

**Table 3 jcm-09-02295-t003:** Overall event outcomes in patients aged <75 and ≥75 years.

	Expected events (×100 patient-years) in ≥75 years	Observed events (×100 patient-years) in ≥75 years	Expected events (×100 patient-years) in <75 years	Observed events (×100 patient-years) in <75 years
Ischemic stroke	7.2 CHADs-VASc score	2.9 Reduction, 59.7% *p* ≤ 0.001	5.1 CHADs-VASc score	1.2 Reduction, 76.5% *p* ≤ 0.001
ICH	1.0 HAS-BLED score	0.3 Reduction, 70.8% *p* = 0.220	0.9 HAS-BLED score	1.2 Increase, 33.3% *p* = 0.642
GI bleeding		6.9		1.5
Major bleeding	6.6 HAS-BLED score (Friberg Registry)	9.0 Increase, 26.7% *p* < 0.001	6.2 HAS-BLED score (Friberg Registry)	3.7 Reduction, 40.3%) *p* = 0.007

GI: gastrointestinal; HT: hypertension; ICH: intracranial hemorrhage.

**Table 4 jcm-09-02295-t004:** Treatment at discharge according to age group.

	<75 (*n* = 326)	≥75 (*n* = 272)	*p*-Value
AAS	208 (63.8%)	166 (61.0%)	0.485
Clopidogrel	207 (63.5%)	164 (60.3%)	0.422
AAS + Clopidogrel	185 (56.7%)	138 (50.7%)	0.142
Anticoagulants (acenocumarol or LMWH)	45 (13.8%)	40 (14.7%)	0.753
NOAC	9 (2.8%)	9 (3.3%)	0.696

AAS: acetylsalicylic acid; LMWH: low molecular weight heparin; NOAC: non-vitamin K antagonist oral anticoagulant.

**Table 5 jcm-09-02295-t005:** Comparison of the main variables in patients presenting GIB events or no GIB events during follow-up.

	GIB (*n* = 35)	No GIB (*n* = 563)	*p*-Value
Age	77.0 ± 8.2	74.0 ± 8.0	0.029
Female	15 (42.9%)	213 (37.8%)	0.553
Hypertension	23 (65.7%)	445 (79.0%)	0.064
Diabetes	14 (40.0%)	190 (33.7%)	0.449
Permanent AF	19 (54.3%)	289 (51.3%)	0.734
Previous stroke	7 (20.0%)	181 (32.1%)	0.133
Previous bleeding	26 (74.3%)	365 (64.8%)	0.254
Previous ICH	4 (11.4%)	156 (27.7%)	0.035
Previous GI bleeding	26 (74.3%)	212 (37.7%)	<0.001
Previous major bleeding	16 (45.7%)	259 (56.0%)	0.973
CHA_2_DS_2_-VASc *	5 [1]	4 [2]	0.390
HAS-BLED *	4 [2]	3 [1]	0.016
Anemia	13 (37.1%)	152 (27.0%)	0.269
Renal failure	10 (28.6%)	117 (20.8%)	0.766
AAS at discharge	23 (65.7%)	352 (62.3%)	0.689
Clopidogrel at discharge	21 (60.0%)	350 (62.2%)	0.798
AAS + Clopidogrel at discharge	19 (54.3%)	304 (54.0%)	0.973
Acenocumarol at discharge	1 (2.9%)	28 (5.0%)	0.572
LMWH at discharge	2 (5.7%)	54 (9.6%)	0.445
NOAC	1 (2.9%)	17 (3.0%)	0.956
Death in follow-up	9 (25.7%)	71 (12.6%)	0.027

AAS: acetylsalicylic acid; AF: atrial fibrillation; GI: gastrointestinal; ICH: intracranial hemorrhage; LMWH: low molecular weight heparin; NOAC: non-vitamin K antagonist oral anticoagulant. * Median (IQR).

**Table 6 jcm-09-02295-t006:** Mean variables of patients with contraindication for OAC treatment reported in different registries.

Registry	EWOLUTION Registry	Multicenter Amplatzer	Amulet Registry	Ii Iberian Registry	Italian Registry
Population (*n*)	*n* = 1025	*n* = 1047	*n* = 1088	*n* = 598	*n* = 613
Mean age	73.4 ± 8.9	75 ± 8	74 ± 8	75.4 ± 8.6	75.1 ± 8.0
Follow up (months)	12	13	12	22.9	20
CHA_2_DS_2_-VASc score (mean ± SD)	4.5 ± 1.6	4.5 ± 1.6	4.5 ± 1.6	4.4 ± 1.5	4.2 ± 1.5
HAS-BLED score (mean ± SD)	2.3 ± 1.2	3.1 ± 1.2	3.3 ± 1.1	3.4 ± 1.2	3.2 ± 1.1
Rate of events per 100 patient-years					
Deaths	9.8%	4.3%	8.4%	7%	7.4%
History of Stroke	30.5%	39%	28%	39%	36.3%
History of major Bleeding	31%	47.7%	72%	46%	41.6%
Observed vs. Expected					
Stroke	1.1% vs. 7.2% (CHA_2_DS_2_-VASc) RRR, 83%	1.8% vs. 5.62% (CHA_2_DS_2_-VASc) RRR, 59%	2.9 vs. 6.7% (CHA_2_DS_2_-VASc) RRR, 57%	1.6% vs. 8.5% (CHA_2_DS_2_-VASc) RRR, 81%	2.9% vs. 8.6% (CHA_2_DS_2_-VASc) RRR, 66%
Major bleeding	2.7% vs. 5% (HAS-BLED) RRR, 46%	2.1% vs. 5.34% (HAS-BLED) RRR, 46%	7.1% vs. 10.3%	3.9% vs. 6.4% (HAS-BLED) RRR, 39%	4.5% vs. 6.3% (HAS-BLED) RRR, 29%

OAC: Oral anticoagulants; RRR: relative risk reduction; SD: standard deviation.

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
