# Peer review of "Major Bleeding Predictors in Patients with Left Atrial Appendage Closure: The Iberian Registry II"

_jcm, 2020, doi:10.3390/jcm9072295_

Round 1

Reviewer 1 Report

Left atrial appendage closure (LAAC) is an alternative to long term oral anticoagulation (OAC) for a small subgroup of patients with atrial fibrillation (AF) and absolute contraindication to OAC or when OAC is ineffective for clot prevention in the left appendage (indication to LAAC plus OAC). However, most of the studies are not conclusive or not enough powered and we still lack evidence of LAAC effectiveness especially in comparison with non-vitamin-K-dependent oral anticoagulation (NOAC).

Two large-scale randomised controlled trials have demonstrated the safety and efficacy of the Watchman device compared to long-term OAC therapy for stroke prevention (ReddyVY et al. JAMA. 2014;312:1988–98 and PREVAIL trial. J Am Coll Cardiol. 2014;64:1–12.). However, in these trials only patients without contra-indications for long-term OAC were included. 

Moreover, we need to understand which is the best post procedure therapy protocol (dual antiplatelet treatment or OAC for short period followed by single antiplatet therapy).

The authors show in a population of 598 patients that age and previous gastrointestinal bleeding identify a population at high risk of bleeding.

This study can make an important contribution in defining patients at risk of bleeding for a possible optimization of post procedural therapy. To this purpose, I would suggest that the authors better define the population and the therapy before and after LAAC.

Main critical issues:

  1. It would be important to describe the study population with more details: 

LAAC indication should be reported

OAC and antiplatelet therapy used before LAAC for stroke prevention Should be report

Did previous bleeding events occur during anticoagulant or antiplatelet therapy? Did GIB events occur with or without anticoagulant/antiplatelet therapy?

was the anamnesis for cancer investigated?

2. Table 5 and Table 4: ASA and Clopidogrel should be considered alone when on mono-therapy. The authors include in the ASA group patients taking ASA only and patients in dual antiplatelet therapy. They did the same for the clopidogrel group.

3. It seems that the risk of bleeding after LAAC is related to therapy, in fact, after first few months there is no difference in bleeding events between the two groups. Therefore, therapy appears more relevant than age or there might be an important therapy/age interaction. I would suggest to analyze the differences by dividing the population into groups according therapy (dual antiplatelets, single antiplatelet or anticoagulant). Moreover, I suggest to analyze events after the first few months (blanking period) when patients start chronic single antiplatelet therapy.

4. In multivariate analysis , does age remain a significant predictor of events even if considered a continuous variable?

Author Response

Response to Reviewer 1 Comments

1 a.- LAAC indication should be reported

1 b. OAC and antiplatelet therapy used before LAAC for stroke prevention Should be report

In order not to add more tables we decide to include this data in text before line 67

LAAC indication was as follows: stroke under OAC therapy 6.2%, previous bleeding 73.7%, high risk of bleeding 14.2% and other (poorly controlled INR, patient decision, etc) 5.9%

And we continue

Before LAAC 74,8% and 25,2% of the patients were under OAC and antiplatelet therapy repectively

1 c. Did previous bleeding events occur during anticoagulant or antiplatelet therapy?

In line 89 we write the data

In patients with previous bleeding, these occurred in 82,1% and 17,9% of patients under OAC and antiplatelet therapy repectively

1 d. Did GIB events occur with or without anticoagulant/antiplatelet therapy?

We clarify this data in line 90

Most of GIB (28 of 35) took place during the first 12 months (25 of them in the first 6 months)

And in line 172

Bleeding events, especially GIB, were higher during the first 6 months post-procedure, which is the time window when a higher percentage of patients were receiving dual antithrombotic treatment (therapy changed from two to one antiplatelet agent during the first 3-6 months) (therapy changed from two antiplatelet agents to only one after the first 3-6 months).

1 e. was the anamnesis for cancer investigated?

Yes it has been added as exclusion criteria in line 67 and also in response to reviewer 2

2. Table 5 and Table 4: ASA and Clopidogrel should be considered alone when on mono-therapy. The authors include in the ASA group patients taking ASA only and patients in dual antiplatelet therapy. They did the same for the clopidogrel group.

Tables 4 and 5 show ASA, Clopidogrel and ASA+Clopidogrel goups

3.- It seems that the risk of bleeding after LAAC is related to therapy, in fact, after first few months there is no difference in bleeding events between the two groups. Therefore, therapy appears more relevant than age or there might be an important therapy/age interaction. I would suggest to analyze the differences by dividing the population into groups according therapy (dual antiplatelets, single antiplatelet or anticoagulant).

Moreover, I suggest to analyze events after the first few months (blanking period) when patients start chronic single antiplatelet therapy.

This is an interesting approach and we comment the limitation of the real life studies in this kind of studies in line 215

It is clear that post-interventional treatment is an important variable to take into account, but in all studies, with the exception of randomized series, patients present a high percentage of bleeding events of different natures and origins. This makes it difficult to standardize treatment guidelines, and doctors must make decisions based on the specific risk associated with each patient.14

Unfortunately, we cannot do a stratified analysis by post-implant therapy as most of the patients were on double therapy during the first 3-6 months

As it has been suggested by the reviewer, a similar analysis based on follow-up time was done in the Registro Ibérico II publication (REC 2019; 72:449-455) where the lower bleeding rate of patients with longer follow-up was highlighted. This data is commented in the first paragraph of the results: “ …… major bleeding events according to CHA2DS2-VASc and HAS-BLED scores in the total population were 1.6% (vs expected 8.5%) and 3.9% (vs expected 6.4%), with a relative risk reduction (RRR) of 81% for stroke and 39% for major bleeding events. In patients monitored for more than 24 months (683 patient-years), stroke and bleeding frequencies were 1.5% and 2.6% with RRRs of 82% and 59%, respectively”.

4. In multivariate analysis, does age remain a significant predictor of events even if considered a continuous variable?

Yes definitely age as a continous variable remains as a major bleeding independent factor after device implantation (HR: 1.2; p<0.001).

Reviewer 2 Report

- scores (such as CHADS-VASC or HAS-BLED) are generally not-normally distreibuted variables. please revise normality criteria for continuous variables (for not-normally distributed variables, please use median and IQR, and non-parametric tests for comparison).

- proportional-hazard assumption should be checked for Cox regression.

- observed / expected events might be confusing. please simplify tables (expected events are generally not reported, please include 95% CI for hazards and P values).

- please describe the rationale for variable inclusion in multivariable regression. baseline differences might account for postoperative differences and should be carefully evaluated in multivariable models.

- results from the univariable analysis should be also presented. considering the number of patients and the number of events, a non-parsimonious approach for univariable analysis should be performed.

- Supplementary Figure: please include number at risk for each group (rather than total), and mark censored cases.

Author Response

Response to Reviewer 2 Comments

Point 1: scores (such as CHADS-VASC or HAS-BLED) are generally not-normally distributed variables. Please revise normality criteria for continuous variables (for not-normally distributed variables, please use median and IQR, and non-parametric tests for comparison).

Changes have been made according to the commentary, in abstract (line 39), and also in Table 1 and Table 5.

Patients aged < 75 or ≥ 75 years had median CHA2DS2-VASc scores of 3.9±1.5 and 5.0±1.4 4 [IQR: 2] and 5 [IQR: 2], respectively (p<0.001) and HAS-BLED scores were 3.3±1.2 and 3.5±1.1 3 [IQR: 1] and 3 [IQR: 1] for each group (p=0.004 0.007).

Point 2: proportional-hazard assumption should be checked for Cox regression.

Proportional-hazard assumption for Cox Regresion was checked by use of Cox proportional hazards regression test with time dependent covariates which established the time-independence for the covariates.

Point 3: observed / expected events might be confusing. please simplify tables (expected events are generally not reported, please include 95% CI for hazards and P values).

The expected incidence of events in the sample was calculated as the mean of the individual risk of each patient. Therefore, each patient was assigned an individual risk factor according to a score of bleeding and ictus risk depending on his or her CHADS and HAS-BLED score, as indicated in the work by Friberg and colleagues in the Swedish Atrial Fibrillation cohort study (we have added this methology in methods section). Therefore we believe it´s important to show those expected incidence of events (see line 73).

“Multivariate analysis (Cox regression) was performed to identify which variables might be independent predictors for bleeding events.

The observed incidence of events (number of events during the follow-up period divided by the number of patients per year of follow-up) was calculated per patient and year of follow-up (number of patients at the beginning of the follow-up period multiplied by the mean time of follow-up of those patients expressed in years).The expected incidence of events in the sample was calculated as the mean of the individual risk for each patient. Each patient was assigned an individual risk according to a score of bleeding and ictus risk depending on his or her CHADS2 and HAS-BLED score, as indicated in the work by Friberg and colleagues in the Swedish Atrial Fibrillation cohort study.

Point 4: please describe the rationale for variable inclusion in multivariable regression. Baseline differences might account for postoperative differences and should be carefully evaluated in multivariable models.

For the multivariate analysis, all those independent variables that were associated with the studied response (dependent variable) with a level of association p <0.2 were selected to be evaluated in the model as covariates. This strategy is validated by multiple studies, as the following:

Mickey RM, Greenland S. The impact of confounder selection criteria on effect estimation. Am J Epidemiol 1989;129:125-37.

Maldonado G, Greenland S. Simulation study of confounder-selection strategies. Am J Epidemiol 1993;138:923-936.

We added the next, in line 74

Statistical analysis

Quantitative variables are expressed as mean (± standard deviation [SD]) or median (interquartile range [IQR]). Categorical variables are expressed as absolute frequency and percentage. Categorical variables were compared using the χ2 test or Fisher's exact test, and quantitative variables using the Student t-test or Wilcoxon test. Comparisons between rates of observed and expected events were evaluated using binomial tests. Event-free survival analysis was performed using the Kaplan-Meier method and Cox regression. Multivariate analysis (Cox regression) was performed to identify which variables might be independent predictors for bleeding events. Proportional-hazard assumption for Cox Regresion was checked by use of Cox proportional hazards regression test with time dependent covariates. All analyses were carried out using the SPSS statistical package, version 19.0.”

Point 5: results from the univariable analysis should be also presented. Considering the number of patients and the number of events, a non-parsimonious approach for univariable analysis should be performed.

We have added the univariate analysis in the text (line 86).

In the univariate analysis, the variables that were associated with a higher rate of “major bleeding events” at follow-up were: age ≥ 75 years (HR: 2.8; CI95%: 1.5-5.2; p = 0.002), gastrointestinal bleeding (GIB) history (HR: 2.3; CI95%: 1.2-4.3; p = 0.007) and the antecedent of hypertension (HR: 0.5; CI95%: 0.3-1; p = 0.047). Multivariate analysis (Cox regression analysis) showed that the variables associated with “major bleeding events” during follow-up were only age ≥ 75 years (HR: 2.5; 95% CI: 1.3 to 4.8; p = 0.004) and gastrointestinal bleeding  GIB history (HR: 2.1; 95% CI: 1.1 to 3.9;  p = 0.020).

Point 6: Supplementary Figure: please include number at risk for each group (rather than total), and mark censored cases.

Supplementary Figure has been modified according to the comment and the new legend:

Figure 2. Cumulative survival free from gastrointestinal bleeding (GIB) is significantly higher in patients < 75 years when compared to patients ≥ 75 years.

Reviewer 3 Report

Please see comments in the attached file.

Line 62, Please specify study type, inclusion and exclusion criteria.

Regarding therapy used? DAPT or warfarin?

Line 127, Move table 3 to the beginning of the next page.

Line 142, Have table 5 start on a new page.

Line 146, What were the baseline characteristics?

Line 172, Not a clear sentence. Please revise

Author Response

Response to Reviewer 3 Comments

  • Line 62, Please specify study type, inclusion and exclusion criteria.

The next paragraph has been included as follows in line 66:

Five hundred and ninety-eight patients from the Iberian Registry II referred for LAAC were recruited from 13 hospitals across the Iberian Peninsula (10 from Spain and 3 from Portugal) between 2 March 2009 and 18 December 2015.5 These were the set of patients prospectively included in the Iberian Registry I who are continuing long-term follow-up, plus additional patients successively included up to the end of the date set for end of recruitment. Inclusion criteria were one or more of the following conditions: serious haemorrhage during anticoagulant therapy, prior disease or clinical event that contraindicated OACs or repeated failure to adequately control INR, and haematologist indication to suspend anticoagulation therapy. Exclusion criteria were malignancy,  life expectancy less than one year and refuse to provide informed consent for this study.

  • Regarding therapy used? DAPT or warfarin?

Treatment at discharge is shown in table 4. Previously most of them were without any antithrombotic therapy due to high risk of bleeding or previous serious bleeding

  • Line 127, Move table 3 to the beginning of the next page.
  • Line 142, Have table 5 start on a new page.

Tables 3 and 5 have been moved to the top of the pages

  • Line 146, What were the baseline characteristics?

The main baseline characteristics are shown in table 1

  • Line 172, Not a clear sentence. Please revise.

We have modified the sentence to clarify

“ therapy changed from two antiplatelet agents to only one after the first 3-6 months”

Round 2

Reviewer 2 Report

.